# Iron Status of Burkinabé Adolescent Girls Predicts Malaria Risk in the Following Rainy Season

**DOI:** 10.3390/nu12051446

**Published:** 2020-05-16

**Authors:** Loretta Brabin, Stephen A. Roberts, Halidou Tinto, Sabine Gies, Salou Diallo, Bernard Brabin

**Affiliations:** 1Division of Population Health, Health Services Research and Primary Care, University of Manchester, Oxford Road, Manchester M13 9PL, UK; steve.roberts@manchester.ac.uk; 2Clinical Research Unit of Nanoro, (IRSS—URCN), B.P.218 Ouagadougou 11, Burkina Faso; halidoutinto@gmail.com (H.T.); saloudiallo89@yahoo.fr (S.D.); 3Department of Biomedical Sciences, Prince Leopold Institute of Tropical Medicine, Antwerp, Belgium and Medical Mission Institute, 97074 Würzburg, Germany; sabine.gies@medmissio.de; 4Liverpool School of Tropical Medicine and Institute of Infection and Global Health, University of Liverpool, Liverpool L7 3EA, UK; b.j.brabin@liverpool.ac.uk

**Keywords:** iron biomarkers, malaria, adolescent girls, menarche, body mass index, Burkina Faso

## Abstract

High levels of storage iron may increase malaria susceptibility. This risk has not been investigated in semi-immune adolescents. We investigated whether baseline iron status of non-pregnant adolescent girls living in a high malaria transmission area in Burkina Faso affected malaria risk during the following rainy season. For this prospective study, we analysed data from an interim safety survey, conducted six months into a randomised iron supplementation trial. We used logistic regression to model the risk of *P. falciparum* infection prevalence by microscopy, the pre-specified interim safety outcome, in relation to iron status, nutritional indicators and menarche assessed at recruitment. The interim survey was attended by 1223 (82%) of 1486 eligible participants, 1084 (89%) of whom were <20 years at baseline and 242 (22%) were pre-menarcheal. At baseline, prevalence of low body iron stores was 10%. At follow-up, 38% of adolescents had predominantly asymptomatic malaria parasitaemias, with no difference by menarcheal status. Higher body iron stores at baseline predicted an increased malaria risk in the following rainy season (OR 1.18 (95% CI 1.05, 1.34, *p* = 0.007) after adjusting for bed net use, age, menarche, and body mass index. We conclude that routine iron supplementation should not be recommended without prior effective malaria control.

## 1. Introduction

Iron deficiency (ID) is a lack of mobilisable iron stores with a compromised supply of iron to tissues [1]. Iron deficiency anaemia (IDA) represents a population sub-group with iron-deficient erythropoiesis. It is estimated that one third of women of child-bearing age will be anaemic, of whom one half will be iron deficient as a result of menses and poor diets [2]. A Sustainable Development Goal for 2025 is a 50% reduction of anaemia in women of reproductive age including non-pregnant (NP) menstruating adolescents [3]. Current guidelines, where anaemia prevalence is >20%, recommend intermittent oral weekly iron and folic acid (IFA) (60 mg iron and 2.8 mg FA) for a period of three months, withdrawal for three months, then recommencement [4]. With anaemia prevalence, >40% daily iron (30–60 mg) is advised for three consecutive months each year. Intermittent supplementation (weekly—once, twice, or more doses) is recommended by the World Health Organization (WHO) in order to reduce side effects and non-adherence associated with daily regimes, although daily iron continues to be recommended during pregnancy [5]. There is evidence, mostly from studies outside of sub-Saharan Africa, of improvements of NP adolescent haemoglobin levels following iron supplementation [6], but in Burkina Faso, our randomised controlled trial showed a lack of efficacy of weekly iron supplementation in both NP and pregnant cohorts [7]. 

IDA contributes to anaemia prevalence, but in much of sub-Saharan Africa, malaria is the more important cause [8]. A meta-analysis, based on national surveys and using adjusted ferritin levels to measure iron, reported a lower than expected ID prevalence (17.8%; 95% CI: 13.4, 22.3) among women of reproductive age in countries with a high burden of infection-drive inflammation [9]. Malaria affects iron metabolism, so malaria-induced destruction of red blood cells hinders reticuloendothelial macrophages in recycling iron back to the bone marrow, leading to anaemia [10]. The relative contribution of ID to the overall burden of anaemia raises critical risk assessment issues in malaria endemic areas, given long-standing concerns that better iron status could increase susceptibility to malaria [10,11] and possibly enteric infections [12,13]. In a prospective Zambian study, children under six years of age with higher iron stores had an elevated malaria risk during the high malaria season [14]. Similarly, in primigravidae, we reported a higher risk of early gestational malaria with increased body iron stores (relative risk 1.53, 95% CI 0.67–2.38, *p* < 0.001) [15]. Cochrane reviews in children [16] and non-adolescent parous menstruating women of child bearing age [4], however, support, and consider it safe to provide IFA supplements in malaria endemic areas provided malaria surveillance and control strategies are in place [17]. The IFA recommendations for menstruating adolescents are nonetheless based on very limited evidence [18,19,20,21].

Improving the nutritional status of adolescents is justifiably one of the key actions highlighted by the Global Accelerated Action for the Health of Adolescents programme [22]. Weekly adolescent IFA programmes have already been introduced in more than ten Asian/African countries [23] and may improve adolescent health in non-malaria endemic regions. The aim of this paper was to examine the effect of adolescent iron status on malaria risk in a high transmission setting in Burkina Faso, one of 21 countries accounting for 85% of global malaria deaths in 2018 [24]. Using iron biomarkers and nutritional data collected at recruitment to a community-based randomised controlled trial of iron supplements [7], we prospectively assessed NP adolescent malaria risk at the pre-specified interim safety survey conducted in the following rainy/malaria season, approximately six months after recruitment. We found that better iron status at the baseline increased subsequent malaria risk.

## 2. Materials and Methods

### 2.1. Ethics Statement

This survey was conducted as part of a randomised, double blind controlled non-inferiority trial conducted in rural Burkina Faso between April 2011 and January 2014. The study received ethical approvals in Burkina Faso; National Ethics Committee (CERS Ref 015-2020/CE-CM) and the Comité d’Ethique pour la Recherche en Santé du Centre Muraz; the United Kingdom Research Ethics Committee, Liverpool School of Tropical Medicine (LSTM/REC protocol 10–55); the Institutional Review Board of the Institute of Tropical Medicine (IRB/AB/AC/016), and the Antwerp University Hospital Ethics Committee, Belgium (EC/UZA). The trial was registered with Clinicaltrials.gov on 27 September 2010: trial registration number NCT01210040. Written, informed consents were given by all individuals, with additional guardian consents provided for minors. The work described was carried out in accordance with the Code of Ethics of the World Medical Association. (Declaration of Helsinki). The main results of the study were communicated to the communities at the end of the study.

### 2.2. Background to the Trial

Recruitment started in April 2011 in thirty rural villages in the Nanoro Health and Demographic Surveillance System catchment area, situated 85 km from Ouagadougou where malaria is hyper-endemic [25]. The entomological inoculation rate in this area is estimated to be 50–60 infective bites/person/year (Diabaté A., personal communication). *Plasmodium falciparum* accounts for nearly all severe malaria cases and almost half of all outpatient consultations. National government malaria control activities are based on diagnosis and treatment of suspected cases, free long lasting insecticidal bed net (LLIN) distribution, and intermittent preventive treatment for pregnant women.

Healthy nulliparous, NP women aged 15–24 years, less than 40% of whom were literate [13], were recruited to receive either weekly ferrous gluconate (60 mg) with folic acid (2.8 mg) as intervention or folic acid alone as the control, following the WHO guidelines updated in 2016 [26]. Participants were visited at home each week by field workers who directly observed ingestion of capsules and monitored for pregnancy and symptoms of illness. In cases of fever (temperature ≥ 37.5 °C) or history of fever in the previous 48 h, the field worker performed a malaria Rapid Diagnostic Test (RDT) and if positive, collected a blood sample for a thick film. RDT positives were referred to the Health Centre and treated at no cost with artesunate-amodiaquine, following national guidelines. The RDT targeted HRP2 antigen (Bioline SD, Malaria Antigen Pf 05FK50) had a demonstrated high detection rate (97.5%) for low parasite density infections and a zero false positive rate on the WHO Product testing round [27]. Bed net use was self-reported as having slept under a net the previous night. Supplement adherence for each woman was computed as the number of weekly treatments received divided by the number of weeks on the trial.

### 2.3. Baseline Assessment

At enrolment, demographic data and medical histories were recorded including history of illness, last menstrual period, and age at menarche, together with clinical examination. Height (nearest mm), weight (nearest 100 g), and mid-upper arm circumference (mm: MUAC) were measured in duplicate. A 5 mL venous blood sample was collected for later iron biomarker assessments. Body Mass Index (BMI) was computed and transformed into an age-specific Z-score based on the 2007 WHO normative standards [28]. Haemoglobin was not measured at the baseline. Women were not recruited if they had clinical signs of severe anaemia (conjunctival or mucosal pallor, tachycardia, respiratory distress), or a history or presence of major clinical disease. HIV prevalence was low (<2%) in the general population. Malaria parasitaemia was not assessed. Symptomatic women were referred to the nearest health centre. All participants received aLLIN) and single doses of albendazole (400 mg) and praziquantel. 

### 2.4. Interim Safety Survey

Following recruitment in the dry season, the interim survey took place in the first malaria season (November/December 2011) after approximately six months supplementation. The purpose was to conduct a blinded interim analysis of safety, based on malaria prevalence in the two trial arms, with pre-specified stopping rules for malaria prevalence and a review of other safety data. All NP participants recruited prior to that date were included (a further cohort of women were recruited to the trial after this date, thus, the numbers here were smaller than those reported elsewhere on the main trial). In addition to the data collected routinely at the weekly visit, a finger prick blood sample for a RDT test and malaria blood smear for microscopy were collected and the previous night’s bed net use was ascertained. Positive RDTs were treated following the national guidelines. Non-attendees were screened at a successive weekly visit during the period of the interim survey, if available.

Following an independent review of this data, the Data Monitoring Committee recommended that the trial should continue as planned and recruitment and follow-up should continue. 

### 2.5. Laboratory Tests

#### 2.5.1. Assessment of Iron Status

At the baseline survey, blood was transported to the research laboratory and within three hours, centrifuged, aliquoted, and stored at −80 °C. Ferritin and serum transferrin receptor (sTfR) were measured using duplicate sampling by Enzyme-linked Immunosorbent Assay (ELISA) (Spectro Ferritin S-22 and TFC 94 Transferrin Receptor, RAMCO Laboratories Inc, Texas) at the Research Laboratory in Nanoro, and C-reactive protein (CRP) by ELISA (EU59131IBL International, GmbH, Hamburg). Intra-assay coefficients of variation were all <10%. Serum hepcidin was measured in the Netherlands by competitive ELISA assay as previously described [29]. Body iron stores (BIS) (mg/kg) were calculated using the equation derived by Cook et al.: body iron (mg/kg) = −[log_10_ (1000 × sTfR/ferritin) − 2.8229]/0.1207 [30]. 

Iron deficiency and iron status were estimated using specific biomarker definitions with adjusted ferritin estimates allowing for inflammation. These were:Unadjusted ferritin <15 μg/L [1] and adjusted ferritin were used. Adjusted ferritin was based on the internal regression correction approach described by Namaste et al. [31], allowing for inflammation as described by Mei et al. [32].A non-pregnant internal regression slope log (ferritin) against log (CRP) estimate was used for ferritin correction, adjusting ferritin levels where CRP exceeded a reference level of the tenth centile to that reference level [15].sTfR concentration of >8.3 µg/mL [33].Ratio of sTfR (mg/L) to log_10_ adjusted ferritin (μg/L) >5.6, which assesses both stored and functional iron and is possibly less affected by inflammation. The ratio derives from the cut-offs sTfR > 8.3 μg/mL and ferritin < 30 μg/L. RAMCO was the assay used for sTfR, as its cut-off (>5.6) best predicted iron deficient bone marrow stores (sensitivity 74%, specificity 73%, accuracy 73%), in another area of high malaria transmission [34].The median hepcidin reference level of serum/plasma based on a healthy non-malarious Dutch population [35]. The hepcidin 95% reference range for women 18–24 years of age from that population was: median 2.6 nM; 2.5th percentile 0.7 nm; and 97.5th percentile 10.5 nM.BIS was calculated using the regression-adjusted ferritin estimate. Low body iron was defined as zero iron stores <0 mg/kg.

#### 2.5.2. Assessment of Malaria

Whole blood for malaria films were Giemsa stained and read by two qualified microscopists. For discrepant findings (positive/negative; >two-fold difference for parasite densities ≥400 μL; >log_10_ if < 400 μL), a third independent reading was made. Parasite density was calculated by counting the number of asexual parasites per 200 white blood cells in the thick blood film by light microscopy at 100× magnification. The parasite density per μL was calculated assuming a white cell count of 8000 mm. Clinical malaria was defined as a positive blood slide with fever (≥37.5 °C) or history of fever.

### 2.6. Statistical Analysis

There was no difference in malaria prevalence risk by trial arm at the interim survey (Table 1, Fisher’s Exact Test: *p* = 0.77). For the remaining interim assessments, data from both trial arms were therefore combined.

Associations of anthropomorphic measures and iron biomarkers with age and menarcheal status were assessed using linear regression models, fitting a common slope for menarcheal and non-menarcheal girls. An interaction term (different slopes) was considered, but not presented as there was no statistical evidence for different slopes, nor was there evidence of non-linear effects. Significance values are presented for the age (slope) and menarche effects. Skewed biomarker values were log-transformed. Mean values and 95% confidence intervals (CI) for each integer age are presented. The relationships between malaria prevalence and age and BMI (Z-score) were similarly analysed using analogous logistic regression models.

Similar logistic regression models were used to quantify the relationships between malaria prevalence at the interim survey and baseline iron biomarkers. A linear, iron effect was fitted for the menarcheal and non-menarcheal groups as there was no statistical evidence for an interaction effect or for any non-linear behaviour. Fitted lines are presented with rug plots to indicate actual data values. Multivariable logistic regression was used to assess the relationship between iron biomarkers and malaria prevalence, after adjusting for age and BMI (expressed as *Z*-score) as linear terms, and bed net use and menarcheal status at the time of assessment as categorical variables. More complex models were considered, but did not improve the fit to the data. Each biomarker was then tested in turn as an addition to this 4-variable base model. Biomarkers were standardised by the mean and standard deviation of the observed distribution in the sample in order to obtain comparable odds-ratio estimates between biomarkers. All analyses were conducted in the R statistical environment (v3.6).

## 3. Results

### 3.1. Sample Description

A total of 1486 NP females were eligible for the interim safety survey, which was completed by 1223 (82%). The number re-screened included 1084 adolescents (<20 years) and 139 young adults (20–24 years) (Table 2). 

The reasons for non-attendance are shown in the participant flow diagram (Figure 1).

Of the adolescents, 22% (242) were non-menarcheal at baseline (Table 2). Mean BMI and MUAC rose with age in both groups, but non-menarcheal had lower values than menarcheal girls (Figure 2).

### 3.2. Iron Biomarker Profiles at the Baseline Survey

Iron deficiency prevalence ranged from 10% with low adjusted BIS (<0 mg/kg) to 26% using the sTfR/log_10_ ferritin >5.6. (Table 2) In Figure 3, values for four iron biomarkers are plotted by age and menarcheal status. Regression slopes for this figure are included in Appendix A. Adjusted ferritin (*p* < 0.001), hepcidin (*p* < 0.001), and adjusted BIS (*p* < 0.001) decreased and the sTfR/log_10_ ferritin ratio (*p* = 0.004) increased with age. The same trends were seen in non-menarcheal girls with very little difference between menarcheal and non-menarcheal girls after adjusting for age.

### 3.3. Malaria Indices at the Interim Safety Survey

Field workers visited participants for a minimum of 12 weeks (median 22, IQR 17–26 weeks). On the night previous to the survey, 76% reported sleeping under a bed net. Malaria blood slides were positive for 38% of adolescents; 53% were RDT positive, with similar proportions for all women (Table 3). Only 4% had clinical malaria. Median parasite density in adolescents was 231 parasites/mm^3^ (IQR 110–656). For all women both RDT (*p* = 0.004) and parasite prevalence *p* < 0.001) decreased with age (Figure 4). Regression slopes for this figure are included in Appendix A. Age-specific RDT positivity or parasite prevalence did not differ between non-menarcheal and menarcheal women. Neither parasite prevalence (*p* = 0.21) nor RDT positivity (*p* = 0.35) decreased significantly with increasing BMI when age-adjusted using Z-scores. 

### 3.4. Baseline Iron Status and Malaria Prevalence at the Interim Survey

A regression analysis, (regression slopes in Appendix A) that included all women, examined the presence of malaria parasitaemia at the interim survey against four baseline iron biomarkers (Figure 5). Parasitaemia prevalence increased with indicators of better iron status, that is, higher adjusted ferritin (*p* = 0.007), BIS (*p* = 0.002) or hepcidin (*p* = 0.049, whereas it decreased with higher sTfR/log_10_ ferritin ratios (*p* = 0.003), indicating poorer iron status. A similar pattern of associations was observed for non-menarcheal girls (Figure 5).

A multivariable analysis was conducted that adjusted for bed net use, age, menarche, and BMI, testing for independent associations with malaria (Table 4). Malaria prevalence assessed by microscopy or RDT decreased independently with increasing age. Increasing baseline BIS was associated with higher risk of malaria (OR 1.18, 95% CI 1.05, 1.34, *p* = 0.007), whereas increasing baseline sTfR/log_10_ ferritin ratios were associated with lower risk (0.84, 95%CI 0.74, 0.96, *p* = 0.007). Bed net use the night previous was associated with RDT positivity, but not microscopy positivity. Menarche and BMI were not independently associated with malaria. 

## 4. Discussion

The interim safety study was conducted in the rainy season to ensure consistently high exposure to malaria. At baseline, during the dry season, between 74–90% of rural, largely illiterate, adolescents had adequate iron stores. In the rainy season survey, more than half had evidence of malaria infections. At a population level, iron deficiency predicted a lower malaria risk six months later, a problematic concept when weighing the risks and benefits of an iron intervention [36]. 

This was the first study to assess the predictive level of iron stores for subsequent malaria risk in a large population of adolescent girls living under hyper-endemic malaria conditions. IFA guidelines for menstruating adolescents were largely based on limited evidence from five (GRADE low) studies (one providing unpublished data) from malaria endemic regions in sub-Saharan Africa [18,19,20,21]. Two were early studies [18,19], some did not discriminate pre- and post-pubertal girls [20], some sample sizes were small (<280) [18,19,21], all but one [18] were school-based and only two reported malaria indices [18,21]. The rationale for routinely providing iron supplements to menstruating adolescents (i.e., adolescents of reproductive age) is mainly based on the assumption that menses increases the risk of iron deficiency anaemia and that there is a need to conserve iron stores before the first pregnancy. Our data showed no difference in iron biomarkers between menstruating and non-menstruating girls after controlling for age (Figure 3). We have also shown that better iron stores actually increased the risk of malaria early in pregnancy [15].

Currently, routine adolescent IFA programmes are considered safe if delivered alongside malaria surveillance and control strategies [17]. In our study, despite receiving impregnated bed nets at enrolment, with weekly active case detection based on febrile episodes and RDT testing, this level of surveillance failed to redress the high prevalence of chronic asymptomatic parasitaemias typical of adolescent malaria. Weekly field worker visits only detected symptomatic cases, and clinical cases were infrequent, as noted elsewhere for adolescents living in stable malaria transmission areas [37]. The reported use of bed nets apparently improved from the baseline, but self-reported net use on the previous night has limited reliability [38]. The effectiveness of both adolescent malaria control and routine iron supplementation in malaria endemic areas are open to question. We reported no reduction in iron deficiency by trial arm in the NP cohort with weekly iron supplementation [7]. Since chronic asymptomatic parasitaemias can limit gut iron absorption [39], reductions in IDA with iron interventions might be minimal without effective malaria control. In Malawian school children, symptomatic management alone made little impact on malaria or anaemia prevalence [40]. In Malian school children aged nine to twelve years, when malaria was reduced through a comprehensive malaria control strategy including provision of LLIN, malaria education, and intermittent preventive treatment, there were significant improvements in haemoglobin [41]. Since many older girls do not attend secondary school, such interventions may be insufficient to reduce re-infections in high transmission areas.

The major strengths of this analysis are the size of the population who attended the interim survey, and the fact that malaria exposure was consistently high, being assessed during a short period in the rainy season. In total, 18% of participants did not attend the interim survey, largely due to temporary absence, as it was not uncommon for young unmarried girls to be sent away to assist other households domestically [42], and then re-join the study on their return. For a prevalence survey, the large sample size was adequate for accurate estimates of malaria and iron indices, although use of rapid tests and microscopy may have underestimated low density (sub-microscopic) parasitaemias. Iron status was assessed comprehensively and controlled for inflammation measured by CRP. Data on anaemia and malaria at baseline were not collected as this was a trial design requirement. At the conclusion of the trial, an end assessment survey for NP women showed that anaemia prevalence (Hb < 12 g/dL) was 48.9% in parasitaemic and 39% in non-parasitaemic women, with a prevalence of ID below 20% [15]. 

Age was an independent predictor of increasing malaria immunity (Table 4), nevertheless many older girls experienced asymptomatic infections and hence were untreated. Iron repletion may favour malaria transmission as these chronically asymptomatic adolescents represent an important reservoir of malaria transmission. In Mali, a country prioritised for high impact malaria control [24], a high proportion of *P. falciparum* infections were concurrently gametocytaemic (51–89%) with a higher prevalence in individuals ≤17 years [43]. In Burkina Faso, gametocyte density relative to total parasite concentration increases with age, also favouring transmission [44]. An area of research that warrants attention is whether body iron status influences gametocyte density. This would require quantifying the reproductive rate of infection relative to the population prevalence of iron deficiency and repletion. The mechanism explaining the predictive utility of the host iron status at the baseline for future malaria risk in the interim survey may relate to both host pathophysiology as well as parasite ontogeny in the mosquito [45].

## 5. Conclusions

National surveys to measure ID and IDA prevalence are required for program planners to estimate the impact and cost-effectiveness of improving iron status [9,46], with specific cut-offs for justifying IFA supplementation. In high malaria transmission areas, the primary goal should be malaria control. Our data also suggest that while iron deficiency cannot be ignored, routine IFA should be delayed until adequate malaria control is ensured. So far, such a strategy has been unacceptable to public health communities [47]. The challenge is to reduce asymptomatic parasitaemias and frequent reinfections and significantly impact this reservoir of malaria transmission before introducing IFA programmes. Malaria control should directly result in reduced anaemia and improved gut absorption of iron, thereby promoting the effectiveness of IFA.

## Figures and Tables

**Figure 1 nutrients-12-01446-f001:**
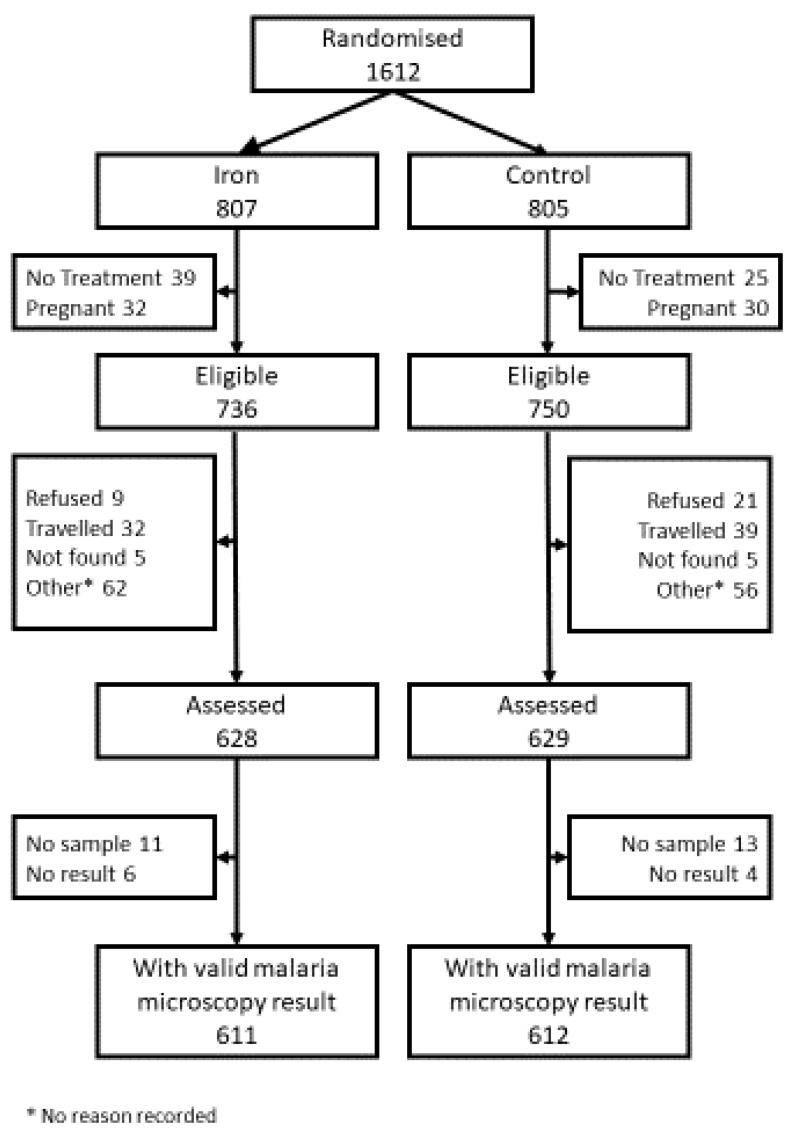
Participant flow.

**Figure 2 nutrients-12-01446-f002:**
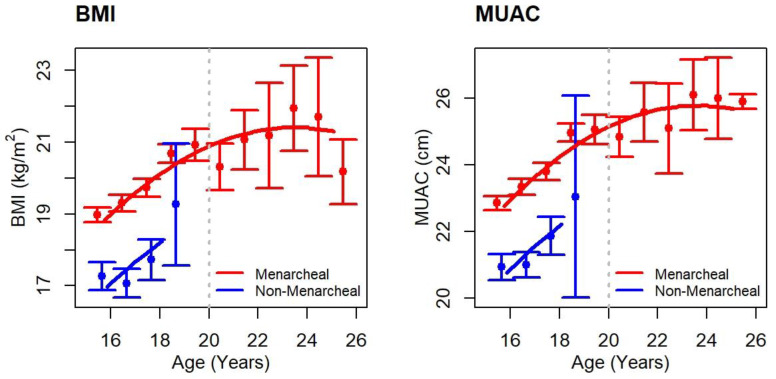
Age specific body mass index (BMI) and mid-upper arm circumference (MUAC) in menarcheal and non-menarcheal participants measured at the baseline. Error bars represent the mean and 95% CI for each age group and are slightly offset for clarity. Red: menarcheal; Blue: non-menarcheal. Lines are a quadratic fit to aid visualisation. Stippled vertical line indicates the adolescent cohort (<20 years).

**Figure 3 nutrients-12-01446-f003:**
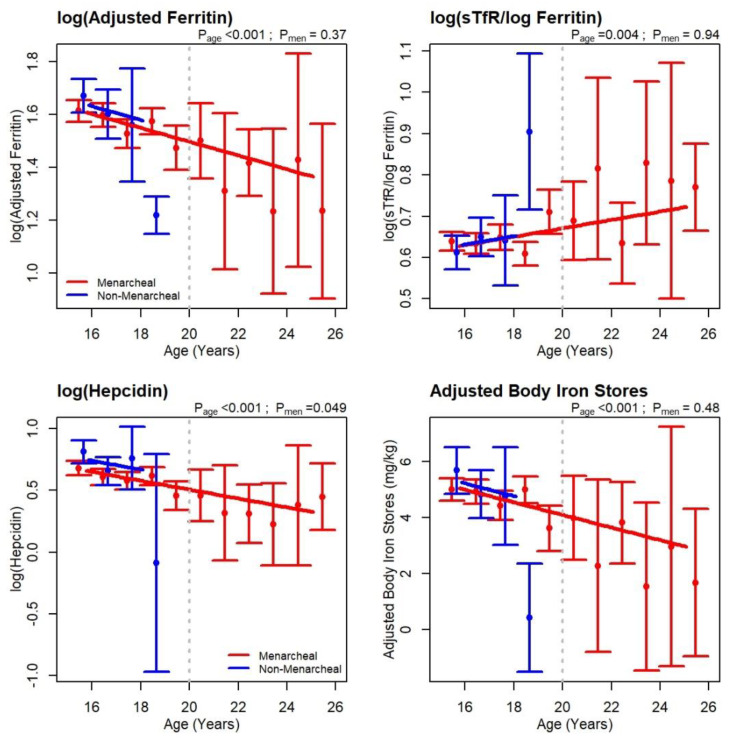
Baseline iron biomarkers by age for menarcheal and non-menarcheal participants. Error bars represent the mean and 95% CI for each age group and are slightly offset for clarity. Red: menarcheal; Blue: non-menarcheal. Lines are linear regression fits assuming a common slope for menarcheal and non-menarcheal girls. Significance levels are shown for the slope (P_age_) and the difference between menarcheal and non-menarcheal (P_men_). Stippled vertical line indicates the adolescent cohort (<20 years). Regression slopes are included in Appendix A.

**Figure 4 nutrients-12-01446-f004:**
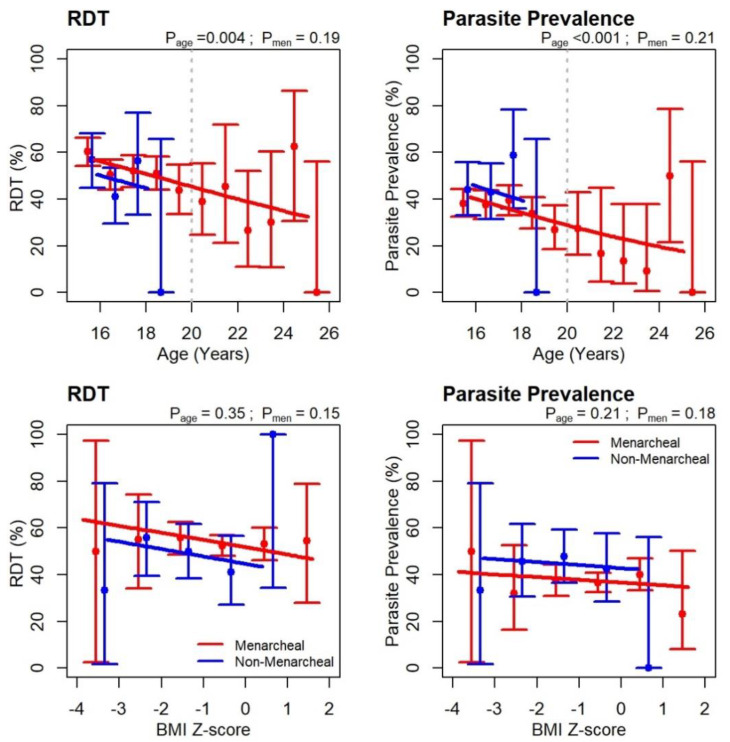
Malaria prevalence by age and BMI *Z*-score at wet season survey. Error bars represent the exact binomial 95% CI for each age or BMI group and are slightly offset for clarity. Red: menarcheal; Blue: non-menarcheal. Lines are logistic regression fits assuming a common slope for menarcheal and non-menarcheal girls. Significance levels are shown for the slope (P_age_) and the difference between menarcheal and non-menarcheal (P_men_). Stippled vertical line indicates the adolescent cohort (<20 years). Regression slopes are included in Appendix A.

**Figure 5 nutrients-12-01446-f005:**
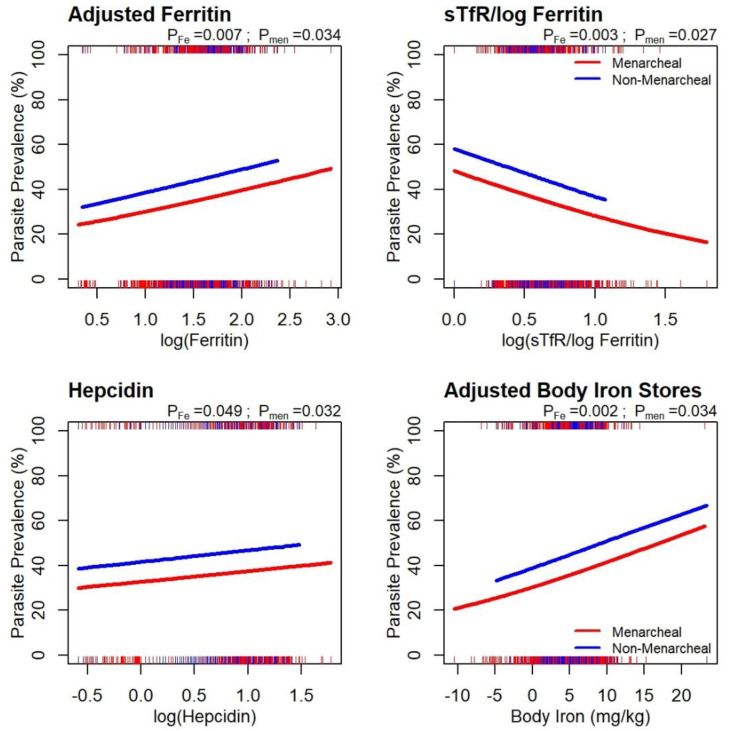
Logistic regression fits for parasite prevalence (microscopy positive) against iron biomarkers. Red: menarcheal; Blue: non-menarcheal. Fitted regression lines based on a cubic-spline fit with a common shape for the two groups. Rugs along the bottom and top axes indicate negative and positive girls, respectively. Significance levels are shown for the slope (P_Fe_) and the difference between menarcheal and non-menarcheal (P_men_). Regression slopes are included in Appendix A.

**Table 1 nutrients-12-01446-t001:** Parasite prevalence by treatment arm at interim survey.

Iron		Controls	
*n*/*N*	% (95% CI)	*n*/*N*	% (95% CI)
222/611	36 (32.0–39.9)	226/612	36.9 (33.2–40.8)

**Table 2 nutrients-12-01446-t002:** Clinical parameters at the baseline and interim wet season assessment in all women and adolescents of the study population.

Clinical Parameters	All Women (15–24 Years)	Adolescents (<20 Years)
**At Baseline**		
Median age, years (IQR)(N)	16 (16–18) (1223)	16 (16–18) (1084)
Menarcheal, *n/N* (%)	978/1223 (80)	842/1084 (78)
Median reproductive age, years (IQR)(N)	2 (1–3) (978)	1 (1–2) (842)
Became pregnant during trial, *n/N*(%)	291/1223 (24)	251/1084 (23)
Median BMI, kg/m^2^ (IQR)(N)	19 (18–21) (1223)	19 (18–21) (1084)
Median BMI *Z* score (IQR)(N)	−0.6 (−1.1–−0.1) (1223)	−0.6 (−1.1–−0.1) (1084)
Median MUAC, cm (IQR)(N)	24 (22–25) (1223)	23 (22–25) (1084)
Bed net use ^a^ n/N(%)	392/1223 (32)	336/1084 (31)
**At Interim Assessment**		
Median age, years (IQR)(N)	17 (16–19) (1223)	17 (16–18) (1084)
Menarcheal, *n/N* (%)	1073/1223 (88)	934/1084 (86)
Median Reproductive age, years (IQR)(N)	2 (1–3) (1073)	1 (1–2) (934)
Bed net use ^a^ *n/N* (%)	914/1198 (76)	813/1061 (77)
Median weeks on supplementation (IQR)(N)	22 (17–26) (1223)	22 (18–26) (1084)
Median % adherence ^b^ (IQR)(N)	69 (56–79) (1223)	69 (56–79) (1084)
**Iron Biomarkers at Baseline**		
Median CRP mg/L (IQR)(N)	0.653 (0.262–1.571) (1216)	0.663 (0.261–1.588) (1078)
Median GM adjusted ferritin, µg/L, (IQR)(N) ^c^	40.4 (22.5–64.2) (1214)	41.2 (24.0–65.3) (1075)
Adjusted ferritin < 15 µg/L, *n/N* (%) ^c^	178/1214 (15)	137/1075 (13)
Median GM sTfR µg/mL, (IQR)(N)	6.46 (5.34–8.01) (1216)	6.44 (5.34–7.94) (1078)
sTfR > 8.3 µg/mL, *n/N* (%)	276/1216 (23)	236/1078 (22)
Median sTfR/log10 ferritin ^c^, (IQR)(N)	4.07 (3.17–5.75) (1213)	4.02 (3.17–5.50) (1075)
sTfR/log_10_ ferritin ^c^ > 5.6, *n/N* (%)	320/1213 (26)	265/1075 (25)
Median hepcidin nmol/L (IQR)(N)	4.8 (2.0–10.4) (1213)	5.0 (2.2–10.7) (1076)
Hepcidin < 0.7 nmol/L, *n/N* (%) ^d^	112/1213 (9)	84/1076 (8)
Median adjusted BIS ^c^, mg/kg (IQR)(N)	5.2 (2.7–7.1) (1213)	5.3 (2.8–7.1) (1075)
Low adjusted BIS < 0 mg/kg, *n/N* (%)	118/1213 (10)	88/1075 (8)

IGR: Interquartile range; BMI: Body mass index; MUAC: Mid-upper arm circumference; GM: geometric mean; BIS: body iron stores; sTfR: serum transferrin receptor; CRP: C-reactive protein; ^a^ Women reporting sleeping under a bed net the night previous to assessment; ^b^ Percentage of directly observed weekly supplements; ^c^ Ferritin and BIS adjusted for inflammation using an internal regression correction (see Methods); ^d^ The 2.5th percentile of the reference range for healthy Dutch women aged 18–24 years.

**Table 3 nutrients-12-01446-t003:** Malaria indices at the interim wet season survey in all women and adolescents of the study population.

Parameter	All Women (15–24 Years)	Adolescents (<20 Years)
Microscopy positive, *n/N* (%) ^a^	446/1223 (36)	414/1084 (38)
Fever, *n/N* (%) ^b^	132/1202 (11)	117/1065 (11)
Clinical malaria, *n/N* (%) ^c^	53/1218 (4)	47/1079 (4)
RDT positive, *n/N* (%) ^d^	605/1177 (51)	551/1044 (53)
Parasite density, parasites/mm^3^, median (IQR)[N]	227 (105–614) (446)	231 (110–656) (414)

^a^*P. falciparum* blood smear positive; ^b^ Temperature ≥ 37.5 °C; ^c^
*P. falciparum* blood smear positive and fever ≥ 37.5 °C; ^d^ RDT: Rapid diagnostic malaria test.

**Table 4 nutrients-12-01446-t004:** Baseline parameters and adjusted risk estimates for malaria at the interim survey.

Parameter	Adjusted Odds Ratio (95% CI)			
	Rapid Test Positivity	*p*-Value	Microscopy Positive	*p*-Value
Bed nets ^a^	1.50 (1.14–1.97)	0.003	1.01 (0.77–1.34)	0.930
Age	0.91 (0.85–0.97)	0.007	0.88 (0.82–0.95)	0.002
Menarche	1.44 (0.97–2.13)	0.069	0.80 (0.54–1.18)	0.260
BMI Z score	0.88 (0.75–1.04)	0.130	0.98 (0.83–1.16)	0.850
Log10 (Adjusted Ferritin)	1.15 (1.02–1.30)	0.021	1.16 (1.03–1.32)	0.016
sTfR	0.87 (0.77–0.98)	0.024	0.87 (0.76–0.99)	0.035
Log10 (sTfR/Log10 ferritin ratio)	0.85 (0.75–0.96)	0.007	0.84 (0.74–0.96)	0.007
Adjusted body iron stores ^b^	1.16 (1.03–1.31)	0.014	1.18 (1.05–1.34)	0.007
Log10 (Hepcidin)	1.13 (1.00–1.27)	0.043	1.10 (0.97–1.24)	0.140

^a^ Slept under bed net on previous night; ^b^ Ferritin and body iron stores adjusted for inflammation using an internal regression correction (see Methods). Logistic regression model adjusting for bed net use, age, menarche, and BMI; Iron biomarkers are standardised by the sample mean and SD and odds ratios expressed per unit on the standardised scale. sTfR: serum transferrin receptor; BMI: Body Mass Index.

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
