# Peer review of "Iron Status of Burkinabé Adolescent Girls Predicts Malaria Risk in the Following Rainy Season"

_nutrients, 2020, doi:10.3390/nu12051446_

Round 1

Reviewer 1 Report

In this manuscript, Brabin et al. describe the relationship between iron status and malaria in a large cohort of young women. The work is well done and well described. Their suggestion to pay attention to the presence and development of malaria infection before treatment with IFA will improve, if taken into consideration, the health of the population in Burkina Faso.

Author Response

No revisions requested.

Reviewer 2 Report

The current study investigated whether iron status affects malaria risk in non-pregnant women of Burkina Faso (a high malaria transmission area). Assessments revealed that higher baseline body iron stores, determined via ferritin, hepcidin, and TfR measurements among others, predicted an increase malaria risk in the following rainy season. The authors concluded that malaria control should be prioritized in high malaria transmission areas, followed by routine iron supplementation when adequate malaria control is in place.

This study was carried out thoroughly in a large sample. The descriptions of the results appear accurate, and the discussion contains correct references to other research. The manuscript was well-prepared, with only minor text edits required.  The authors sufficiently addressed this reviewers only comment - the lack of hemoglobin levels at baseline – in the discussion.

Author Response

This reviewer did not specify any revisions.

Reviewer 3 Report

The authors have obtained interesting results about the relations between iron status of adolescent girls and malaria risk in a high transmission setting in Burkina Faso. The authors conclude malaria control should directly result in reduced anaemia and improved gut absorption of iron, thereby promoting the effectiveness of intermittent oral weekly iron and folic acid.

Please consider only this suggestion:

Results

  • The author should consider include:

Tables 2 and 3 titles … in all women and adolescents of the study population.

Tables 2 and 3 row one,

Clinical Parameters

All women

(20-24 years)

Adolescents

(<20 years)

  • Figure 1 is very explanatory.

Author Response

  1. The headings to Tables 2 and 3 have been expanded to include the words "..in all women and adolescents of the study population" (page 5, lines 204-5; page 8, lines 246-7).
  2. Age categories are added as descriptors (15-24 years) and <20 years) at the head of the columns of the same tables (page 5, Table 2; page 8, Table 3).